# Lipotrichaibol A and Trichoderpeptides A–D: Five New Peptaibiotics from a Sponge-Derived *Trichoderma* sp. GXIMD 01001

**DOI:** 10.3390/md23070264

**Published:** 2025-06-24

**Authors:** Weichan Yang, Zhenzhou Tang, Xiaowei Luo, Yuman Gan, Meng Bai, Houwen Lin, Chenghai Gao, Ling Chai, Xiao Lin

**Affiliations:** 1Guangxi Key Laboratory of Marine Drugs, Institute of Marine Drugs, Guangxi University of Chinese Medicine, Nanning 530200, China; 15278363299@139.com (W.Y.); trcstrive2015@126.com (Z.T.); luoxiaowei1991@126.com (X.L.); gan_ym2018@163.com (Y.G.); xxbai2014@163.com (M.B.); gaoch@gxtcmu.edu.cn (C.G.); 2Research Center for Marine Drugs, Department of Pharmacy, Ren Ji Hospital, Shanghai Jiao Tong University School of Medicine, Shanghai 200127, China; franklin67@126.com; 3Guangxi Key Laboratory of Traditional Chinese Medicine Quality Standards, Guangxi Institute of Chinese Medicine and Pharmaceutical Science, Nanning 530022, China

**Keywords:** sponge-associated fungi, peptaibiotics, *Trichoderma* sp., cytotoxic activity

## Abstract

Five previously undescribed peptaibiotics, including one 7-mer lipopeptaibol named lipotrichaibol A (**1**), and four 11-mer peptaibiotics named trichoderpeptides A-D (**2**–**5**) were isolated from the rice culture medium of the sponge-derived fungus *Trichoderma* sp. GXIMD 01001. Their structures and absolute configurations were unambiguously established by extensive spectroscopic data analysis and advanced Marfey’s method. All isolated compounds were evaluated via CCK8 bioassays to investigate their antiproliferative activity. Only compound **1** exerted potent cytotoxicity against HT-29 and DLD-1 cells with IC_50_ values at 10.3 ± 1.9 and 12.31 ± 1.5 μM, respectively. In further in vitro bioassay, compound **1** exhibited significant inhibition in colony formation assay, induced apoptosis and blocked the cell cycle in the G0/G1 phase. The mechanism may be related to the regulation of the Erk1/2 signaling pathway.

## 1. Introduction

Natural peptides have attracted remarkable attention over the past decades due to their advantageous characteristics, such as high affinity, specificity, and low toxicity. These peptides have demonstrated significant potential in a myriad of therapeutic areas, especially in cancer treatment [1,2]. Filamentous fungi are among the most abundant and prolific producers of natural peptides [3]. Within this group, *Trichoderma* species are notable for producing diverse bioactive peptides, including a distinct class known as peptaibiotics [4,5,6]. Marine-derived *Trichoderma* have been extensively explored for their pharmaceutical and agrochemical potential. Over the years, numerous secondary metabolites, including terpenoids, polyketides, peptides, alkaloids, and steroids, have been characterized from this species. Peptides, particularly peptaibiotics, represent the second largest class from this genus after terpenes, with 131 compounds reported since 1993 [7].

Peptaibiotics are linear peptides composed of 4–21 amino acid residues. They are distinguished by a high proportion of α,α-dialkylated amino acid residues, such as 2-aminoisobutyric acid (Aib) and isovaline (Iva) [4]. Based on their chemical structures, peptaibiotics have been classified into two major categories: (i) peptaibols, which contain an acylated *N*-terminus and an amide-bound amino alcohol at the *C*-terminus, representing the largest subgroup of peptaibiotics; (ii) non-peptaibol peptaibiotics, including lipoaminopeptides, lipopeptaibols, and other related compounds containing Aib/Iva residues [8,9]. Peptaibiotics exhibit a variety of bioactivities, including anticancer, anti-inflammatory, antimicrobial effects, etc. [10,11]. Extensive prior research has revealed that the mechanism of action of peptaibiotics is primarily associated with their ability to form pores in the cellular membrane [10,12].

In our previous investigations of *Trichoderma* sp. GXIMD 01001, we isolated a series of 18-residue peptaibols named trichorzins A–G. These peptaibols are characterized by an acylated *N*-terminus and either a tryptophanol or phenylalaninol at the C-terminus. They exhibited significant cytotoxicity against four human carcinoma cell lines, with IC_50_ values ranging from 0.46 to 4.7 μM [13]. Inspired by these findings, we expanded our chemical exploration of this strain, leading to the discovery of five new non-peptaibol peptaibiotics, including one lipopeptaibol named lipotrichaibol A (**1**) and four 11-mer peptaibiotics with a free C-terminal Leu residue, namely trichoderpeptides A-D (**2**–**5**). Herein, the details on isolation, structure elucidation, and cytotoxicity of **1**–**5** were reported.

## 2. Results

### 2.1. Isolation and Structural Elucidation

A peptide-enriched subfraction extracted from the rice medium of *Trichoderma* sp. GXIMD 01001 was analyzed by MS and revealed the presence of an unusual MS profile with [M + H]^+^ ions ranging from *m/z* 844 to 1083 (Appendix A). The extract was separated using a series of chromatographic techniques, including normal-phase and reversed-phase medium-pressure liquid chromatography (MPLC), as well as reversed-phase HPLC, to yield five peptaibiotics **1**–**5** (Figure 1).

Compound **1** was obtained as a brown amorphous solid and assigned the molecular formula C_45_H_77_N_7_O_8_ based on HRESIMS (*m/z* 844.5904 [M + H]^+^). The protonated ion of **1** in the tandem mass experiment produced a series of b- and y-type daughter ions. The appearance of b-type ions at *m/z* 727.4746 (b_6_), 614.3912 (b_5_), 543.3547 (b_4_), 444.2862 (b_3_), 331.2021 (b_2_), and 184.1346 (b_1_) provided successive neutral loss of Leuol^7^, Leu^6^, Ala^5^, Iva^4^, Leu^3^, and Phe^2^ from the C-terminus (Figure 2), respectively, establishing the partial amino acid sequences Phe^2^-Leu^3^-Iva^4^-Ala^5^-Leu^6^-Leuol^7^. The fragment ion y_1_ at *m/z* 118.1240 also supported the presence of Leuol^7^ (leucinol) at the C-terminus. The ^1^H NMR spectrum (Table 1, Appendix A) showed seven NH protons in the region of *δ*_H_ 6.8–8.4, five aromatic protons at *δ*_H_ 7.19–7.26, five α-H protons at *δ*_H_ 3.6–4.6 and a long acyl chain (CH_2_)_n_ moiety at *δ*_H_ 1.25, which are characteristic signals for lipopeptides [8,14,15]. The ^13^C and DEPT135 NMR data (Appendix A) revealed the presence of 45 carbons assigned as seven carbonyls, two quaternary carbons, thirteen methines, thirteen methylenes, and 10 methyl carbons. The COSY, HMBC, HSQC, and TOCSY experiments of **1** (Appendix A) produced spin systems that were easily assigned for an *n*-Oct (*n*-Octanoyl) chain, one Gly (glycine), one Phe (Phenylalanine), one Iva, one Ala (Alanine), one Leuol (Leucinol), and two Leu (Leucine) residues. The NH (*δ*_H_ 8.15) and α-CH_2_ (*δ*_H_ 3.73 and 3.52) of Gly showed NOESY correlations with α-CH_2_ (*δ*_H_ 2.10) of the octanoyl group, suggesting that the *n*-Oct chain tethered with Gly. Thus, the first amino acid residue at the N-terminus should be Gly. The HMBC correlations from the NH protons to the two flanking carbonyls, as well as the NOESY correlations between NH protons of the two adjoining residues, helped in the establishment of the complete sequence of **1** as Octanoyl-Gly^1^-Phe^2^-Leu^3^-Iva^4^-Ala^5^-Leu^6^-Leuol^7^, designated lipotrichaibol A.

Compound **2** was isolated as a white amorphous solid. The molecular formula of **2** was determined to be C_46_H_80_N_12_O_15_ based on the HRESIMS (*m/z* 1041.5914 [M + H]^+^) (Figure 3), accounting for 13 degrees of unsaturation. The ^1^H NMR data (Table 2) exhibited thirteen amide protons in the range *δ*_H_ 6.50–9.00, seven *α*-H at *δ*_H_ 3.65–4.25, and numerous overlapping proton signals between 1.47 and 1.60 ppm comprising 12 methyl signals. The ^13^C and DEPT NMR data revealed the presence of 46 carbons assigned as thirteen carbonyls, five quaternary carbons, seven methines, five methylenes, and sixteen methyl carbons. Interpretation of the ^1^H-^1^H COSY, HSQC, and HMBC data (Appendix A) established eleven amino acid residues, including one Ser (Serine), one Ala (Alanine), one Gln (Glutamine), one Val (Valine), one Gly (Glycine), five Aib (α-Aminoisobutyric acid), along with an Ac (Acetyl) moiety at the N-terminus, and one Leu (Leucine) residue at the C-terminus. The key NOESY signals (Figure 4), which began from the methyl protons (*δ*_H_ 1.90) at the Ac moiety to the subsequent amide protons NH-1 (*δ*_H_ 8.64, Aib^1^)/NH-2 (*δ*_H_ 8.18, Ser^2^), NH-4 (*δ*_H_ 7.83, Aib^4^)/NH-5 (*δ*_H_ 7.55, Aib^5^), NH-6 (*δ*_H_ 7.90, Gln^6^)/NH-7 (*δ*_H_ 7.68, Aib^7^)/NH-8 (*δ*_H_ 6.93, Val^8^)/NH-9 (*δ*_H_ 7.56, Aib^9^), NH-10 (*δ*_H_ 7.82, Gly^10^)/NH-11 (*δ*_H_ 7.48, Leu^11^), helped to string up the full sequence of **2** as Ac-Aib^1^-Ser^2^-Ala^3^-Aib^4^-Aib^5^-Gln^6^-Aib^7^-Val^8^-Aib^9^-Gly^10^-Leu^11^. The sequence of **2** was also supported by the ESIMS^2^ fragments for the protonated ion at *m/z* 1041.5914 [M + H]^+^. The daughter ions (Figure 3) at *m/z* 286.1398, 371.1921, 456.2451, 584.3033, 669.3566, 768.4248, 853.4774, and 910.4990 are in accordance with the above-proposed sequence for **2**, designated trichoderpeptide A.

Compound **3** was isolated as a white amorphous solid and assigned the molecular formula C_47_H_82_N_12_O_15_ based on its HRESIMS data (*m/z* 1055.6108 [M + H]^+^). Careful analysis of 1D and 2D NMR data (Appendix A and Figure 4) revealed its planar structural similarity to compound **2**, except for one amino acid residue: the Aib^5^ in **2** was replaced by Iva^5^ in **3**, which was supported by the appearance of *β*-methylene protons at *δ*_H_ 2.17 and 1.62 and a multiplet *β*-methyl at *δ*_H_ 0.70 instead of two geminal *α*-methyl groups in **2**. The HMBC correlations (Appendix A), from the NH at *δ*_H_ 7.49 of the Iva^5^ to the carbonyl of the Aib^4^ at *δ*_C_ 174.0 and the NH at *δ*_H_ 7.81 of the Gln^6^ to the carbonyl group of Iva^5^ at *δ*_C_ 174.1 confirmed the Iva residue occupied at the position 5. The ESIMS^2^ fragmentation analysis also verified this change, suggesting the linear sequence of **3**, named trichoderpeptide B, as Ac-Aib^1^-Ser^2^-Ala^3^-Aib^4^-Iva^5^-Gln^6^-Aib^7^-Val^8^-Aib^9^-Gly^10^-Leu^11^.

Compound **4** was obtained as a white amorphous solid. The molecular formula was determined to be C_48_H_84_N_12_O_15_ on the basis of HRESIMS (*m/z* 1069.6263 [M + H]^+^). Interpretation of 1D and 2D NMR spectra (Figure 4 and Appendix A) revealed a single variation in the amino acid residue compared to compound **3**, where Aib^4^ was replaced with Iva^4^ in **4**. The disappearance of two *β*-methyl signals in **3**, along with new *β*-CH_2_ protons at *δ*_H_ 2.20 and 1.94 and a *β*-CH_3_ at *δ*_H_ 1.33, supported this variation (Appendix A). The amino acid sequence, including the position of the new Iva, was further corroborated by HMBC correlations in addition to MS^2^ analysis. Thus, compound **4** was identified as Ac-Aib^1^-Ser^2^-Ala^3^-Iva^4^-Iva^5^-Gln^6^-Aib^7^-Val^8^-Aib^9^-Gly^10^-Leu^11^ and named trichoderpeptide C.

Compound **5** was obtained as a white amorphous solid. On the basis of HRESIMS studies, the molecular formula C_49_H_86_N_12_O_15_ was deduced from the [M + H]^+^ at *m/z* 1083.6414. Analysis of MS and 1D NMR (Appendix A) suggested that compound **5** contains one additional Iva instead of Aib in its sequence, compared to compound **4**. Subsequent analysis of HMBC and NOESY correlations (Figure 4), along with neutral loss patterns observed in the MS^2^ experiment (Appendix A), confirmed this hypothesis. These findings enabled the determination of the amino acid sequence of compound **5** as Ac-Aib^1^-Ser^2^-Ala^3^-Iva ^4^-Iva^5^-Gln^6^-Iva^7^-Val^8^-Aib^9^-Gly^10^-Leu^11^, which was named trichoderpeptide D.

The absolute configurations of the amino acid residues in compounds **1**–**5** were established by UPLC-MS analysis of their acid hydrolysates derivatized with Marfey’s method [16]. The results indicated the presence of L-configuration for Phe, Ala, Ser, Val, Leu, Glu (the result of Gln hydrolysis in Marfey ‘s analysis), and Leuol, whereas the Iva was determined as the D-configuration in compounds **1** and **3**–**5**. Additionally, electron circular dichroism (ECD) spectra of **1**–**5** show two negative maximum peaks near 207 and 225 nm (Appendix A), supporting right-handed helical conformations.

### 2.2. Biological Activity Assessment

The cytotoxic effects of all isolated compounds were evaluated using the Cell Counting Kit-8 (CCK-8) assay against three human colorectal cancer cell lines, including HT-29, DLD-1, and SW620. Among them, compound **1** demonstrated a notable inhibitory effect on HT-29 and DLD-1, with IC_50_ values at 10.3 ± 1.9 μM and 12.3 ± 1.5 μM, respectively, outperforming the positive control, cisplatin (Table 3). To further investigate the antiproliferative properties of compound **1**, clonogenic assays and cell cycle analyses were conducted. As depicted in Figure 5A, the number of colony-forming cells was markedly reduced in the drug-treated groups compared to the untreated control in both HT-29 and DLD-1 cells. Flow cytometric analysis further revealed that after 24 h of exposure to compound **1**, there was a significant increase in the proportion of cells in the G0/G1 phase, accompanied by a corresponding decrease in the S phase population (Figure 5B). These findings suggest that compound **1** exerts its antiproliferative activity, at least partially, through the induction of cell cycle arrest at the G0/G1 phase.

To further validate the antiproliferative effects of **1** against HT-29 and DLD-1 cells, Calcein acetoxymethyl ester/propidium iodide (Calcein-AM/PI) staining assay was performed to assess cell viability and morphology. As demonstrated in Figure 6A, treatment with **1** resulted in a pronounced reduction in cell proliferation in both cell lines, accompanied by an increasing number of dead cells, as indicated by red fluorescence staining. These results confirm the antiproliferative activity of compound **1**, consistent with the results from the CCK-8 assay, clonogenic assay, and cell cycle analysis.

Recently, an increasing number of studies have demonstrated that, in addition to membrane-targeted disruption, intracellular mechanisms are also involved in the antitumor activity of peptaibiotics. For instance, the 22-mer gichigamin A was found to exhibit cytotoxic activity through mitochondrial damage [17]. Likewise, Zhang et al. reported that Trichokonin VI, a 20-mer peptaibiotic, inhibited the growth of A549 cancer cells by triggering mitochondrial apoptosis and activating pathways related to Bax and Bak proteins, which are involved in apoptosis and autophagy, respectively [18]. Given that compound **1** exhibited a potential inhibitory effect against human colorectal cancer cells HT-29 and DLD-1, we next investigated its underlying mechanism of action. Specifically, we focused on the extracellular signal-regulated kinase (ERK) pathway, a key branch of the MAPK signaling cascade critical for cell proliferation [19]. Western blot analysis revealed that treatment with compound **1** significantly reduced the levels of phosphorylated ERK1/2 (p-ERK) in both HT-29 and DLD-1 cells (Figure 6B), indicating ERK signaling suppression. Combined with our earlier observations of G0/G1 phase arrest and increased cell death, these findings suggested the ERK pathway inhibition as a likely contributor to the antiproliferative effect of compound **1**.

### 2.3. ADMET Properties

An ADMET study would be a valuable addition to strengthen the biological evaluation. In silico prediction of ADMET properties was conducted using ADMETlab 3.0. As shown in Table 4, Compound **1** exhibited excellent intestinal absorption, with a higher human intestinal absorption (HIA) value of 1.0, indicating favorable bioavailability. The predicted volume of distribution (VD) was 0.455 L/kg, which falls within the normal range (0.04–20 L/kg), suggesting balanced tissue distribution. In terms of metabolic interaction, while the CYP2C19 inhibition probability was 0.81 (suggesting potential drug interaction risk through this pathway), the inhibition probabilities for CYP1A2, CYP2C9, and CYP2D6 were all below 0.5, and compound **1** was not predicted to be a substrate of any major CYP enzymes, demonstrating favorable metabolic stability. Regarding elimination, compound **1** showed a moderate clearance rate at 5.44 mL/min/kg and a relatively short half-life (T_1_/_2_) of 0.486 h, suggesting **1** could be rapidly eliminated from the body. In terms of toxicity, compound **1** exhibited low genotoxicity (AMES = 0.05) and negligible risks of carcinogenicity or rat oral acute toxicity. However, it was predicted to exhibit potential hepatotoxicity (H-HT = 0.676). Overall, compound **1** demonstrates promising pharmacokinetic and pharmacodynamic profiles, though the potential for CYP2C19-mediated interactions and hepatotoxicity warrants further experimental validation.

## 3. Materials and Methods

### 3.1. General Experimental Procedures

HRESIMS spectra were obtained on a Waters Xevo G2 Quadrupole Time of Flight Mass Spectrometry with ACQUITY UPLC BEH C18 column (Waters, Milford, MA, USA, 100 × 2.1 mm i.d.; 1.7 μm). Semipreparative HPLC was performed on an LC2030 platform (Shimadzu, Kyoto, Japan) coupled with a DAD detector using an XBridge BEH Shield RP18 column (Waters Corporation, Milford, Ireland, 10 × 250 mm, 5 μm). NMR spectra were measured on AV-500 MHz, AV-600 MHz, or ASENND TM 800 MHz NMR spectrometers (Bruker Corporation, Billerica, MA, USA). CD spectra were measured on a JASCO J-1500 spectropolarimeter (JASCO, Easton, PA, Tokyo, Japan).

### 3.2. Extraction and Separation of Crude Extracts

The fungus *Trichoderma* sp. GXIMD 01001 was isolated from a sponge collected in the Beibu Gulf, Guangxi, in October 2020. Identification was based on morphological characteristics and ITS rDNA sequence data, with the GenBank accession number OP935742.

The strain extract (59.8 g) was obtained by large-scale fermentation in our previous report, and fifteen subfractions were obtained [13]. All of the crude extracts were further analyzed by UPLC-HRMS: compound **1** with [M + H]^+^ ions at *m/z* 844 were found in the ninth subfraction, and compounds **2**–**5** with [M + H]^+^ ions ranging from *m/z* 1040 to 1083 were found in the twelfth subfraction (Fr.B9). Fr.B9 (9.0 g) was further separated by ODS (from 40% MeCN/60% H_2_O to 100% MeCN) to obtain 9 different fractions (Fr.B9B1–Fr.B9B9). Fr.B9B8 was separated by semi-preparative high-performance liquid chromatography (30% aqueous MeCN with 0.1% formic acid, flow rate at 2 mL/min) to obtain compound **1** (*t*_R_ = 20.0 min, 244 mg). Fr.B12B2 (229.9 mg) was separated by Sephadex LH-20 column chromatography and semi-preparative HPLC (37% aqueous MeCN with 0.1% formic acid) to obtain compound **2** (*t*_R_ = 30.1 min, 10.6 mg). Fr.12B3 (534.2 mg) was subjected to semi-preparative HPLC (46% aqueous MeCN with 0.1% formic acid) to obtain compound **3** (*t*_R_ = 10.5 min, 5 mg). Fr.12B4 (150 mg) was separated by Sephadex LH-20 column chromatography and semi-preparative high-performance liquid chromatography (38% aqueous MeCN with 0.1% formic acid) to obtain compound **4** (*t*_R_ = 21.5 min, 9.8 mg). Fr.12B5 (95.1 mg) was separated by Sephadex LH-20 column chromatography and semi-preparative HPLC (41% aqueous MeCN with 0.1% formic acid) to obtain compound **5** (*t*_R_ = 32.0 min, 6 mg).

Lipotrichaibol A (**1**). Brown amorphous solid; ${\left[\alpha \right]}_{D}^{20}$ + 17.8 (c 0.2, MeOH); NMR data, Table 1; HRESIMS *m/z* 844.5904 [M + H]^+^ (*calcd*. for C_45_H_77_N_7_O_8_, 844.5912, △ 0.95 ppm).

Trichoderpeptide A (**2**). White amorphous solid; ${\left[\alpha \right]}_{D}^{20}$ + 4.1 (c 0.2, MeOH); NMR data, Appendix A; HRESIMS *m/z* 1041.5914 [M + H]^+^ (*calcd*. for C_45_H_77_N_7_O_8_, 1041.5944, △ 2.88 ppm).

Trichoderpeptide B (**3**). White amorphous solid; ${\left[\alpha \right]}_{D}^{20}$ + 9.2 (c 0.2, MeOH); NMR data, Appendix A; HRESIMS *m/z* 1055.6108 [M + H]^+^ (*calcd*. for C_45_H_77_N_7_O_8_, 1055.6101, △ 0.66 ppm).

Trichoderpeptide C (**4**). White amorphous solid; ${\left[\alpha \right]}_{D}^{20}$ + 2.9 (c 0.2, MeOH); NMR data, Appendix A; HRESIMS *m/z* 1069.6262 [M + H]^+^ (*calcd*. for C_45_H_77_N_7_O_8_, 1069.6257, △ 0.47 ppm).

Trichoderpeptide D (**5**). Yellow amorphous solid; ${\left[\alpha \right]}_{D}^{20}$ + 3.2 (c 0.2, MeOH); NMR data, Appendix A; HRESIMS *m/z* 1083.6464 [M + H]^+^ (*calcd*. for C_45_H_77_N_7_O_8_, 1083.6414, △ 4.61 ppm).

### 3.3. Marfey’s Analysis

The sample hydrolysis and derivatization procedures were based on our previously published method [13]. The absolute configurations of each compound were determined by comparing the retention times of the amino acid derivatives with those of the standards.

### 3.4. ECD Analysis

Compounds **1**–**5** dissolved in MeOH were prepared at a 0.25 mg/mL concentration. The spectra were recorded from 400 to 200 with 100 nm/min scanning speed at the response time of 1 s and 2 nm bandwidth.

### 3.5. CCK8 Assay

The cytotoxicity of compounds **1**–**5** against human colorectal cancer cell lines (HT-29, DLD-1, and SW620; all obtained from the Chinese Academy of Sciences, Shanghai, China) was evaluated in vitro using the Cell Counting Kit-8 (CCK-8) assay. Briefly, the test compounds were dissolved in DMSO and subsequently diluted with culture medium to prepare a series of concentrations. Cells were harvested using trypsin, seeded into 96-well microtiter plates (5 × 10^3^ cells per well), and incubated in either RPMI-1640 medium (6124574, Gibco) (for HT-29 and DLD-1) or L-15 medium (2307001, Solarbio) (for SW620) at 37 °C in a humidified atmosphere containing 5% CO_2_ for 24 h. Following this pre-incubation, cells were treated with the serially diluted compounds for 48 h. Cell viability was then assessed using the CCK-8 (KeyGEN, Nanjing, China) assay according to the manufacturer’s instructions, and absorbance at 450 nm was measured with a microplate reader (Bio Tek Instruments, Bad Friedrichshall, Germany). Cisplatin (purity ≥ 98%, Sigma, St. Louis, MO, USA) and 0.01% DMSO served as positive and negative controls, respectively. All experiments were performed in triplicate, and IC_50_ values were calculated using GraphPad Prism 10.0 software.

### 3.6. Cell Clone Formation Experiment

For colony formation assays [20], DLD-1 and HT-29 cells were seeded into 6-well plates at a density of 800 cells per well. After incubation at 37 °C for 24 h, the medium was replaced with fresh medium containing the indicated concentrations of compound **1** (0, 9, or 18 μM). The culture medium was refreshed every four days. After two weeks of continuous culture, colonies were fixed with 4% paraformaldehyde for 15 min and subsequently stained with crystal violet solution (Yuanye, Shanghai, China) for 20 min. Colony formation was quantified using ImageJ 1.54g software.

### 3.7. Calcein-AM/PI Staining Assay

For the Calcein-AM/PI staining assay, DLD-1 and HT-29 cells were seeded in 24-well plates at 5 × 10^4^ cells per well. After 24 h of culture, the 1640 medium was replaced with a medium containing compound **1** (18 μM). After incubation for 0, 6, 12, and 18 h, cells were stained by using the calcein AM (MedChem Express (Monmouth Junction, NJ, USA), 5 μM) and propidium iodide (MedChem Express, 50 μg/mL). The cells were incubated in the dark at 37 °C for 30 min, followed by imaging with an inverted fluorescence microscope.

### 3.8. Cell Cycle Experiment

To assess the effect of compound **1** on cell cycle progression, DLD-1 and HT-29 cells were seeded into 6-well plates at a density of 4 × 10^5^ cells per well and cultured for 24 h. Cells were then treated with compound **1** at the indicated concentrations (0, 9, and 18 μM) for an additional 24 h. Following treatment, cells were collected, washed, and fixed in 75% pre-chilled ethanol at −20 °C for 24 h. After fixation, cells were centrifuged, washed twice with cold PBS, and stained with propidium iodide (PI, 0.1 mg/mL) in the presence of RNase A (0.1 mg/mL) at 37 °C for 30 min. Cell cycle distribution was analyzed by flow cytometry (LSRFortessa, BD, Canton, MA, USA). To enhance the clarity of the results, cell populations were selectively gated using the ModFit LT 5.0 software.

### 3.9. Western Blotting Analysis

Western blotting was performed as previously described [21]. HT-29 and DLD-1 cells were treated with compound **1** (0, 9, and 18 μM) at 37 °C for 24 h. Cells were washed twice with PBS and lysed in RIPA buffer (Beyotime Biotechnology, Jiangsu, China) on ice for 30 min. Lysates were centrifuged at 12,000 rpm for 20 min at 4 °C, and the supernatant containing total protein was collected. Equal amounts of protein were separated by SDS-PAGE and transferred onto polyvinylidene fluoride (PVDF) membranes. Membranes were blocked with 5% non-fat milk at room temperature for 1 h, followed by incubation with the appropriate primary and secondary antibodies. Protein bands were visualized using the Omni-ECL Femto Light Chemiluminescence Kit (Epizyme, Shanghai, China) and imaged with the Amersham Imager 680 (Cytiva, Marlborough, MA, USA).

### 3.10. Prediction of ADMET

In silico predictions of ADMET properties were conducted using the web-based ADMET prediction platform ADMETlab 3.0 (https://admetlab3.scbdd.com/, accessed on 16 June 2025). The SMILES structure of compound **1** was entered into the ADMET Evaluation module.

### 3.11. Statistical Analysis

Statistical analysis was performed using one-way analysis of variance (ANOVA). For data that did not meet the assumptions of normality, the Mann–Whitney U test was applied. Statistical significance was denoted as follows: * *p* < 0.05, ** *p* < 0.01, and *** *p* < 0.001.

## 4. Conclusions

In conclusion, we have identified five new peptaibiotics from a sponge-symbiotic fungus, including one lipopeptaibol named lipotrichaibol A (**1**), comprising seven amino acids with *n*-Oct chain moiety, and four linear peptaibiotics, named trichoderpeptides A-D (**2**–**5**), each composed of eleven amino acids. Although numerous peptaibiotics have been discovered over decades, the five compounds reported in this study exhibit distinct structural features compared to previously documented analogues. Compound **1** is notably characterized by a Phe residue at position 2, whereas most known lipopeptaibols typically possess a Gly or Ala at the same position [14,22]. In addition, compounds **2**–**5** represent the first 11-mer peptaibiotics with a free C-terminal carboxylic acid group. Previously reported peptaibiotics bearing a free C-terminus generally have longer peptide chains (16, 18, 19, or 21 amino acid residues) [6] and were mostly inferred through mass spectrometry rather than direct isolation and structural elucidation.

The cytotoxicity of compounds **1**–**5** was evaluated against three human colon cancer cell lines, with only compound **1** demonstrating significant inhibition. This finding suggests the importance of the hydrophobic property of peptaibiotics in their anticancer activity, particularly compound **1**, which possesses a long-chain acyl group at the N-terminus, and other peptaibols like trichorzins A–G with more amino acid residues, as previously reported [14]. Further investigation revealed that compound **1** effectively induces apoptosis in human colorectal cancer cell lines HT-29 and DLD-1. The observed antiproliferation activity may be attributed, at least in part, to its inhibitory effects on the Erk1/2 signaling pathway, which disrupts the progression of the cancer cell cycle process and consequently suppresses tumor cell proliferation. In silico prediction of ADMET properties indicates that compound **1** exhibits favorable pharmacokinetic properties, highlighting its attractive drug-like potential.

## Figures and Tables

**Figure 1 marinedrugs-23-00264-f001:**
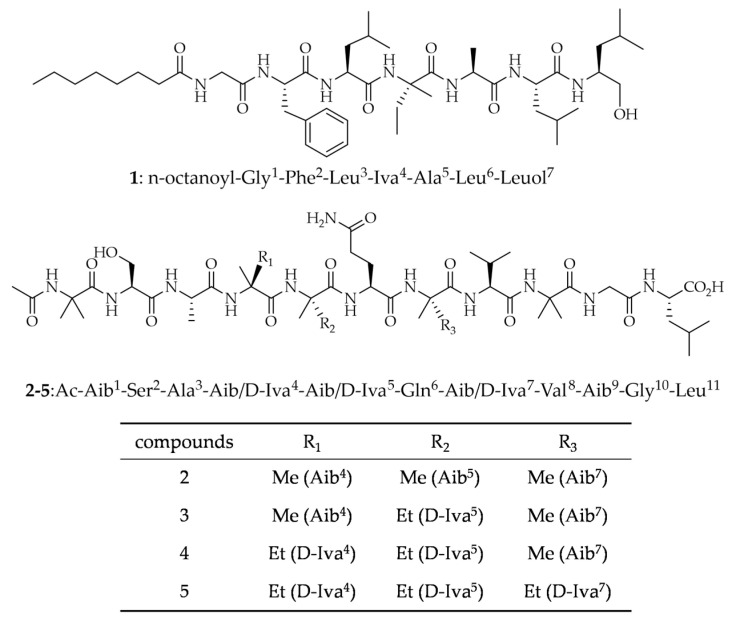
Structures of compounds **1**–**5.**

**Figure 2 marinedrugs-23-00264-f002:**
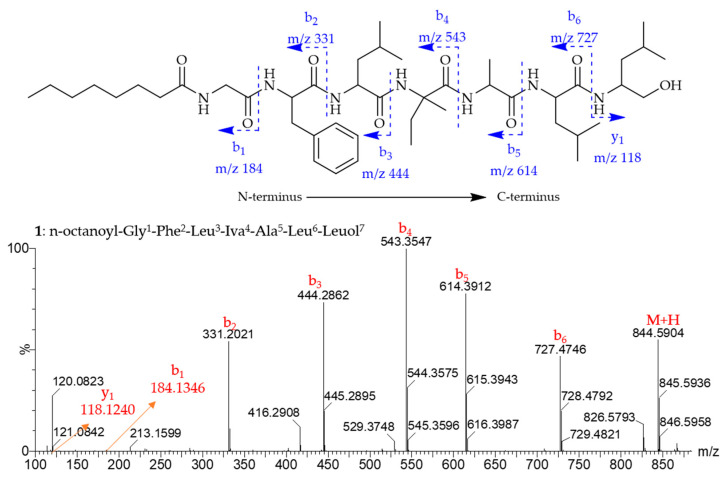
Selected (+) HRESIMS^2^ fragments of compound **1** and MS^2^ fragmentation pattern of the protonated molecular ion [M + H]^+^ (*m/z* 844.5904).

**Figure 3 marinedrugs-23-00264-f003:**
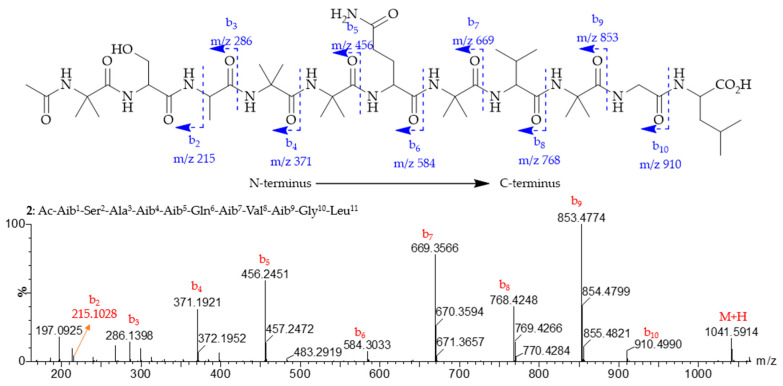
Selected (+) HRESIMS^2^ fragments of compound **2** and MS^2^ fragmentation pattern of the protonated molecular ion [M + H]^+^ (*m/z* 1041.5914).

**Figure 4 marinedrugs-23-00264-f004:**
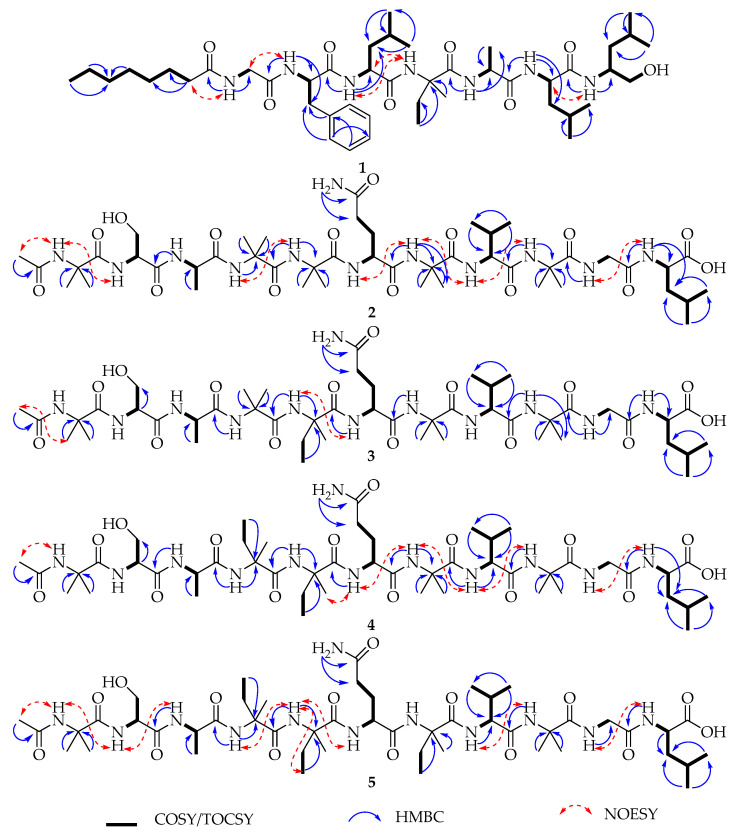
Key COSY/TOCSY, HMBC, and NOESY correlations of compounds **1–5**.

**Figure 5 marinedrugs-23-00264-f005:**
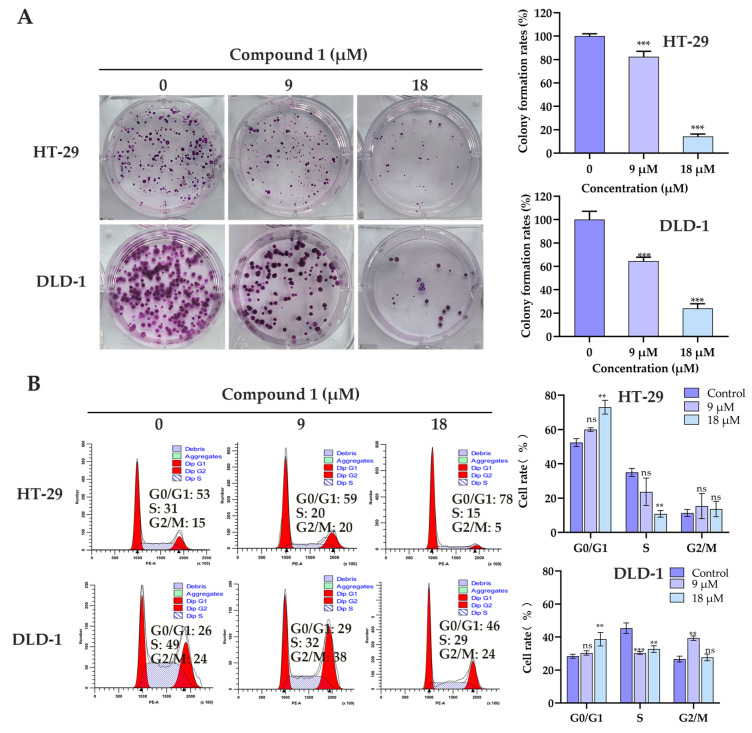
The effect of compound **1** on the proliferation and colony growth of colorectal cancer cells. (**A**) Colony formation assay for detecting cell proliferation of HT-29 and DLD-1 cells treated with compound **1**. Results are expressed as mean ± SD (*n* = 3). ***, *p* < 0.001. (**B**) Flow cytometry of cell cycle in HT-29 and DLD-1 cells treated with 9 and 18 μM compound **1** for 24 h. ns, not significant, ** *p* < 0.01, ***, *p* < 0.001, compared to the control group.

**Figure 6 marinedrugs-23-00264-f006:**
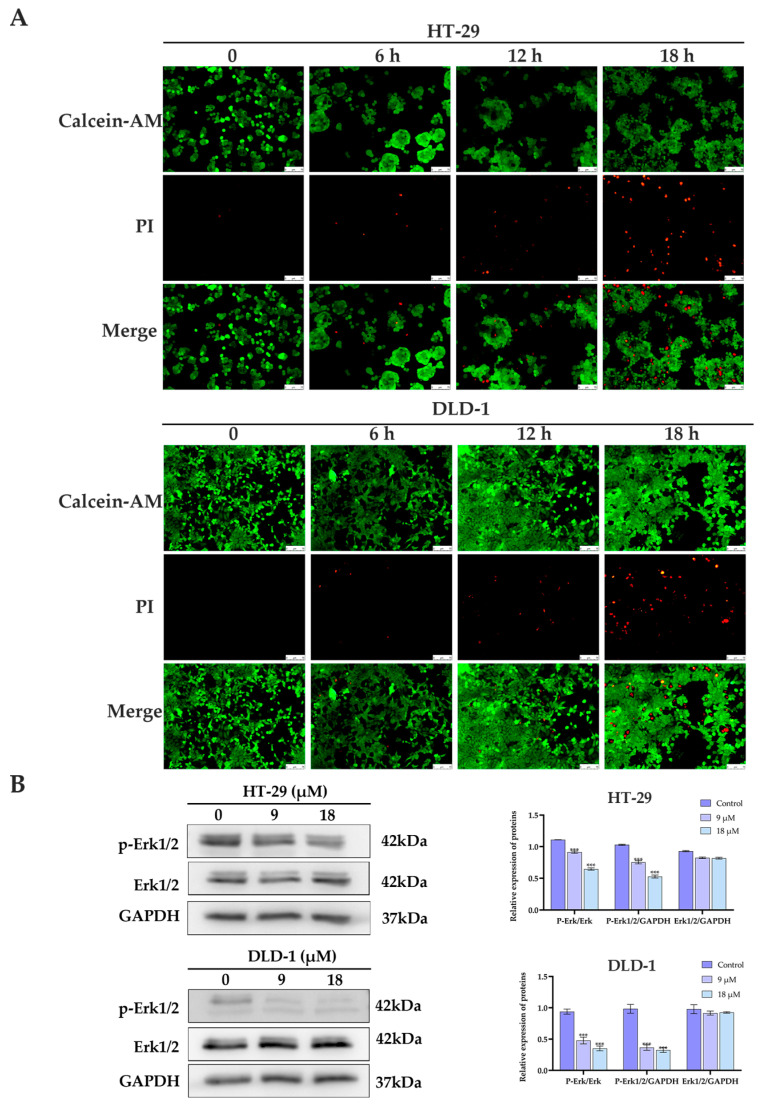
Compound 1-induced cell death. (**A**) Fluorescence images of live–dead staining of HT-29 and DLD-1 cells treated with 18 μM compound **1**, respectively. The bar is 75 μm. (**B**) The regulation of compound **1** on the signaling pathways of HT-29 and DLD-1 Erk1/2 in colorectal cancer cells. *** *p* < 0.001, compared with the control group.

**Table 1 marinedrugs-23-00264-t001:** ^1^H (500 MHz) and ^13^C (125 MHz) NMR data of compound **1** in DMSO-*d*_6_.

Pos.	*δ*_C_, Type	*δ*_H_, (*J* in Hz)	Pos.	*δ*_C_, Type	*δ*_H_, (*J* in Hz)	Pos.	*δ*_C_, Type	*δ*_H_, (*J* in Hz)
*n*-Oct			^3^Leu			^6^Leu		
1	173.2, C		1	172.9, C		1	171.5, C	
2	35.0, CH_2_	2.10 *^a^*, m	2	51.8, CH	4.07 *^f^*, m	2	53.1, CH	4.06 *^f^*, m
3	25.0, CH_2_	1.47 *^b^*, m	3a	40.1, CH_2_	1.33, m	3a	39.2, CH_2_	1.61 *^g^* m
4	28.6, CH_2_	1.23 *^c^*, m	3b		1.25 *^c^*, m	3b		1.53, m
5	28.5, CH_2_	1.23 *^c^*, m	4	24.3, CH	1.63 *^g^*, m	4	24.0, CH	1.56, m
6	31.1, CH_2_	1.22 *^c^*, m	5	22.7, CH_3_	0.91, d, (6.1)	5	23.2, CH_3_	0.87, d, (6.4)
7	22.1, CH_2_	1.23 *^c^*, m	6	21.6, CH_3_	0.85, m	6	21.5, CH_3_	0.76, d, (6.5)
8	13.9, CH_3_	0.85, m	NH		7.53, m	NH		8.05, d, (6.3)
^1^Gly			^4^Iva			^7^Leuol		
1	169.9, C		1	175.1, C		1	48.4, CH	3.79, m
2a	42.4, CH_2_	3.73, dd, (16.3, 5.6)	2	58.7, C		2a	39.9, CH_2_	1.61 *^g^*, m
2b		3.52, dd, (16.3, 5.4)	3a	26.6, CH_2_	2.05 *^a^*, m	2b		1.53, m
NH		8.15, t, (5.6)	3b		1.71, m	3	24.1, CH	1.61 *^g^*, m
			4	7.3, CH_3_	0.71, t, (7.5)	4	23.4, CH_3_	0.80, m
			5	22.2, CH_3_	1.28 *^c^*, m	5	21.2, CH_3_	0.78, s
			NH		7.90, s	6a	64.1, CH_2_	3.26, m
						6b		3.14, m
						NH		6.88, d, (9.2)
						OH		4.48 *^e^*, m
^2^Phe			^5^Ala					
1	172.3 *^d^*, C		1	172.3 *^d^*, C				
2	54.5, CH	4.48 *^e^*, m	2	50.0, CH	3.96, m			
3a	54.5, CH_2_	3.06, dd, (14.0, 4.2)	3	17.1, CH_3_	1.27 *^c^*, m			
3b		2.81, dd, (14.0, 9.7)	NH		7.52, m			
1′	137.7, C							
2′,6′	129.1, CH	7.26, m						
3′,5′	128.1, CH	7.24, m						
4′	126.4, CH	7.19, m						
NH		8.22, d, (7.5)						

*^a–g^* Assignments for overlapping ^1^H and ^13^C NMR resonances with the same superscript may be interchanged.

**Table 2 marinedrugs-23-00264-t002:** ^1^H (600 MHz) and ^13^C (150 MHz) NMR data of compound **2** in DMSO-*d*_6_.

Pos.	*δ*_C_, Type	*δ*_H_, (*J* in Hz)	Pos.	*δ*_C_, Type	*δ*_H_, (*J* in Hz)	Pos.	*δ*_C_, Type	*δ*_H_, (*J* in Hz)
Ac-^1^Aib			^5^Aib			^9^Aib		
COCH_3_	171.1 *^a^*, C		1	174.1, C		1	173.5, C	
COCH_3_	22.8 *^b^*, CH_3_	1.90, s	2	55.8 *^c^*, C		2	56.1 *^c^*, C	
1	175.7, C		3	26.7 *^d^*, CH_3_	1.36 *^h^*, m	3	23.0 *^d^*, CH_3_	1.37 *^h^*, m
2	55.8 *^c^*, C		4	25.9 *^e^*, CH_3_	1.40 *^h^*, m	4	25.9 *^e^*, CH_3_	1.43 *^h^*, m
3	24.0 *^d^*, CH_3_	1.30 *^h^*, m	NH		7.55, s	NH		7.56, s
4	26.1 *^e^*, CH_3_	1.40 *^h^*, m						
NH		8.64, br s						
^2^Ser			^6^Gln			^10^Gly		
1	171.5, C		1	173.3, C		1	168.6, C	
2	57.6 *^f^*, C	4.06, m	2	56.1 *^c^*, C	3.77, m	2a	42.5, CH_2_	3.57, m
3a	60.5, CH_2_	3.68, m	3	26.0, CH_2_	2.00, m	2b		
3b		3.74, m	4a	31.4, CH_2_	2.18, m	NH		7.82, m
OH		5.36, m	4b		2.28, m			
NH		8.18, br s	5	173.5, C				
			6-NH_2_a		7.20, s			
			6-NH_2_b		6.75, s			
			NH		7.90, s			
^3^Ala			^7^Aib			^11^Leu		
1	174.3, C		1	174.6, C		1	170.9, C	
2	50.7 *^g^*, C	4.05, m	2	56.2 *^c^*, C		2	50.8 *^g^*, C	4.04, m
3	16.1, CH_3_	1.30, m	3	23.1 *^d^*, CH_3_	1.37 *^h^*, m	3a	40, CH_2_	1.47, m
NH		7.89, s	4	26.4 *^e^*, CH_3_	1.44 *^h^*, m	3b		1.62, m
			NH		7.68, s	4	24.1, CH	1.67, m
						5	21.5, CH_3_	0.82, d, (6.5)
						6	22.8 *^b^*, CH_3_	0.87, d, (6.6)
						NH		7.48, d, (7.7)
^4^Aib			^8^Val					
1	174.8, C		1	176.8 *^a^*, C				
2	55.8 *^c^*, C		2	57.6 *^f^*, C	4.12, m			
3	23.3 *^d^*, CH_3_	1.35 *^h^*, m	3	28.5, CH	2.27, m			
4	26.1 *^e^*, CH_3_	1.43 *^h^*, m	4	17.6, CH_3_	0.84, d, (6.8)			
NH		7.83, s	5	19.4, CH_3_	0.80, d, (6.9)			
			NH		6.93, d, (9.1)			

*^a–h^* Assignments for overlapping ^1^H and ^13^C NMR resonances with the same superscript may be interchanged.

**Table 3 marinedrugs-23-00264-t003:** IC_50_ values of compound **1**–**5** for all tested cancer cells (*n* = 3).

Compounds	Cytotoxicity, IC_50_ (μM)		
	HT-29	DLD-1	SW620
**1**	10.3 ± 1.9	12.3 ± 1.5	20.0 ± 0.6
**2**–**5**	>40	>40	>40
cisplatin	18.3 ± 0.5	14.1 ± 0.4	15.2 ± 1.2

**Table 4 marinedrugs-23-00264-t004:** Predicted ADME parameters of compound **1.**

Properties	Parameters	Predicted Values
Absorption	HIA (Human Intestinal Absorption)	1.0
Distribution	VD (Volume Distribution)	0.455 L/kg
Metabolism	CYP1A2 inhibitor	0.0
	CYP1A2 substrate	0.0
	CYP2C19 inhibitor	0.64
	CYP2C19 substrate	0.0
	CYP2C9 inhibitor	0.0
	CYP2C9 substrate	0.0
	CYP2D6 inhibitor	0.0
	CYP2D6 substrate	0.0
Elimination	T 1/2 (Half-Life Time)	0.486 h
	CL (Clearance Rate)	5.44 mL/min/kg
Toxicity	hERG (hERG Blockers)	0.196
	H-HT (Human Hepatotoxicity)	0.676
	AMES (Ames Mutagenicity)	0.05
	DILI (Drug-Induced Liver Injury)	0.06
	ROAT (Rat Oral Acute Toxicity)	0.113
	Carcinogenicity	0.011

## Data Availability

The data presented in this study are available upon request from the corresponding author.

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
