# Peer review of "Lipotrichaibol A and Trichoderpeptides A–D: Five New Peptaibiotics from a Sponge-Derived *Trichoderma* sp. GXIMD 01001"

_marinedrugs, 2025, doi:10.3390/md23070264_

Round 1
Reviewer 1 Report
Comments and Suggestions for Authors
The manuscript needs a minor revision by addressing following issues:
- Title of the manuscript should be modified to fit the content and results.
- The introduction should provide more details about the previously reported groups of metabolites and the major biological activities of marine-derived Trichoderma species, particularly those from sponges.
- Page 2, line 56: Please describe how the peptides-enriched subfractions were prepared.
- Please re-check and revise the multiplicities of the protons presented in the Tables of the NMR data to avoid incorrect assignments. For examples: protons of the benzene ring, methyl groups, etc…
- MS/MS fragmentation and NMR data for compound 2 should be shown in the main manuscript.
- HMBC correlations of all the methyl groups should be presented in Fig. 3.
Author Response
|
Comments 1: Title of the manuscript should be modified to fit the content and results. |
|
Response 1: We appreciate the reviewer’s observation regarding the alignment of the title with the manuscript content. Following your suggestion, we have revised the title to more accurately reflect our findings and discussions. The updated title is included in the manuscript.
|
|
Comments 2: The introduction should provide more details about the previously reported groups of metabolites and the major biological activities of marine-derived Trichoderma species, particularly those from sponges. |
|
Response 2: We thank the reviewer for this insightful suggestion. In response, we have enriched the introduction with additional information on previously reported marine-derived Trichoderma metabolites and their significant biological activities. The revised sections are marked in red on Page 2, lines 34-38 of the manuscript.
Comments 3: Page 2, line 56: Please describe how the peptides-enriched subfractions were prepared. Response 3: We appreciate the reviewer’s request for clarification regarding the preparation of peptide-enriched subfractions. In adherence to your suggestion, we have included a detailed description of this preparation method on Page 2, lines 51-54, adding clarity to the experimental procedure.
Comments 4: Please re-check and revise the multiplicities of the protons presented in the Tables of the NMR data to avoid incorrect assignments. For examples: protons of the benzene ring, methyl groups, etc Response 4: Thank you for meticulously pointing out this critical aspect. We have reviewed the NMR data tables with heightened scrutiny and corrected the proton multiplicities, ensuring accurate assignments. All changes were marked in red in the revised manuscript.
Comments 5: MS/MS fragmentation and NMR data for compound 2 should be shown in the main manuscript. Response In line with your recommendation, we have incorporated the MS/MS fragmentation and NMR data for compound 2 into the main body of the manuscript. This additional data is presented in Figure 3 and Table 2, providing a comprehensive overview.
Comments 6: HMBC correlations of all the methyl groups should be presented in Fig. 3. Response 6: We thank the reviewer for this valuable suggestion. The HMBC correlations for all methyl groups have now been fully incorporated into Figure 4. This enhancement provides clarity and supports the structural elucidation. |

Reviewer 2 Report
Comments and Suggestions for Authors
Yang et al. present the isolation and structural identification of new five peptaibiotics as a continuation of their work reported in ref. 14.
These peptaibiotics were isolated from Trichoderma sp. GXIMD 01001 and one of them showed anticancer properties against colorectal HT-29 and DLD-1 cells. Also a proposed mechanism for the anticancer activity was investigated.
While the study explores an interesting class of compounds, the techniques are appropriate and correctly applied, the results are well presented, there are still some issues to be solved before it can be considered for publication.
In introduction, more data about what this manuscript add new comparing to the data presented in reference 14.
No explanation about the choice of colorectal cancer cells to test the isolated peptaibiotics.
No data about the selectivity of compound 1 against cancer cells over normal cells. This is crucial for the overall potential of these compounds.
No ADMET properties for compound 1 are given and discussed. An ADMET study, even at a theoretical level, would be a valuable addition to strengthen the biological evaluation.
Minor issues:
There are some typing errors/spaces to be corrected in the manuscript (e.g. “peptaiboitics”, “such as2-amino…” etc)
In Table 1 there a dd signal without coupling constant values.
Author Response
Reply to reviewer 2
Comments 1: No explanation about the choice of colorectal cancer cells to test the isolated peptaibiotics.
Response 1: We appreciate your inquiry about the rationale for selecting colorectal cancer cells for testing. Colorectal cancer presents high incidence and mortality rates, making it a critical focus in our research on anti-tumor natural products. Our laboratory is dedicated to exploring the effects and mechanisms of anti-colorectal cancer agents, with promising preliminary results such as the notable activity of compound 844 against colorectal cancer cells, which showed lower IC50 values compared to other cancer cell lines. This supports our focus on colorectal cancer cells.
Comments 2: No data about the selectivity of compound 1 against cancer cells over normal cells. This is crucial for the overall potential of these compounds.
Response 2: Thank you for requesting additional information regarding the selectivity of compound 1. We have conducted further experiments to assess its cytotoxicity in both normal colon cells (NCM460) and cancer cell lines (HT-29 and DLD-1). The results indicate that compound 1 has an IC50 value of 17.86 ± 0.4 μM for normal cells.These findings suggest that compound 1 exhibits limited selectivity for cancer cells over normal cells.
Comments 3: No ADMET properties for compound 1 are given and discussed. An ADMET study, even at a theoretical level, would be a valuable addition to strengthen the biological evaluation.
Response 3: We appreciate your suggestion concerning the ADMET properties of compound 1. In response, we conducted an analysis of the ADMET characteristics and included the findings in the manuscript. These additional insights, positioned in Table 4, Section 2.3 on ADMET properties, and Section 3.10 on ADMET prediction, reinforce our biological evaluation and understanding of the compound’s therapeutic efficacy.
Comments 4: There are some typing errors/spaces to be corrected in the manuscript (e.g. “peptaiboitics”, “such as2-amino…” etc)
Response 4: Thank you for your diligent review. We have undertaken a thorough revision of the manuscript to correct typographical errors and spacing inconsistencies, ensuring accuracy and coherence throughout the document.
Comments 5: In Table 1 there a dd signal without coupling constant values.
Response 5: We sincerely apologize for this oversight and appreciate your attentiveness. Upon re-examination of the H-NMR data for compound 1, we identified that the δH 3.96 peak for 2-CH of 5Ala was erroneously marked as dd instead of m. This correction has been applied, evidenced in the revised Table 1 in the resubmitted manuscript.
